

# Engineering a tunable bicistronic TetR autoregulation expression system in *Gluconobacter oxydans*

Monica Bertucci, Ky Ariano, Meg Zumsteg and Paul Schweiger

Department of Microbiology, University of Wisconsin-La Crosse, La Crosse, WI, United States

## ABSTRACT

Acetic acid bacteria are well-known for their ability to incompletely oxidize their carbon sources. Many of the products of these oxidations find industrial uses. Metabolic engineering of acetic acid bacteria would improve production efficiency and yield by allowing controllable gene expression. However, the molecular tools necessary for regulating gene expression have only recently started being explored. To this end the ability of the activation-dependent $P_{lux}$ system and two constitutive repression $P_{tet}$ systems were examined for their ability to modulate gene expression in *Gluconobacter oxydans*. The activation-dependent $P_{lux}$ system increased gene expression approximately 5-fold regardless of the strength of the constitutive promoter used to express the *luxR* transcriptional activator. The $P_{tet}$ system was tunable and had a nearly 20-fold induction when the *tetR* gene was expressed from the strong constitutive promoters $P_{0169}$ and $P_{264}$, but only had a 4-fold induction when a weak constitutive promoter ($P_{452}$) was used for *tetR* expression. However, the $P_{tet}$ system was somewhat leaky when uninduced. To mitigate this background activity, a bicistronic TetR expression system was constructed. Based on molecular modeling, this system is predicted to have low background activity when not induced with anhydrotetracycline. The bicistronic system was inducible up to >3,000-fold and was highly tunable with almost no background expression when uninduced, making this bicistronic system potentially useful for engineering *G. oxydans* and possibly other acetic acid bacteria. These expression systems add to the newly growing repertoire of suitable regulatable promoter systems in acetic acid bacteria.

## INTRODUCTION

Acetic acid bacteria (AAB) are a unique group of organisms that incompletely oxidize their carbon and energy substrates (*e.g.* sugars, polyols, alcohols). This incomplete oxidation is done using a set of membrane-bound dehydrogenases that are oriented toward the periplasmic space and release their products into the medium (*Deppenmeier & Ehrenreich, 2009*; *Prust et al., 2005*). This feature has been widely used in the production of vinegar, vitamin C, 6-amino-L-sorbose to produce the antidiabetic drug miglitol, tanning agents (dihydroxyactone and erythrulose), and flavor and fragrance products (*Deppenmeier, Hoffmeister & Prust, 2002*; *Wang et al., 2016*).

Corresponding author
Paul Schweiger,
pschweiger@uwlax.edu

The ability to express target genes in acetic acid bacteria is of recent interest, especially in the industrially important genera *Gluconobacter, Gluconacetobacter, Komagataeibacter, Acetobacter*, and *Acidiphilium*. Gene expression allows basic research into the physiology and metabolism of these bacteria, as well as for genetic and metabolic engineering. To date most engineering in AAB has been done using constitutive promoters (*Blank & Schweiger, 2018*; *Kallnik et al., 2010*; *Kiefler, Bringer & Bott, 2015*; *Kosciow et al., 2014*; *Merfort et al., 2006*; *Meyer, Schweiger & Deppenmeier, 2013*; *Siemen et al., 2018*; *Yuan et al., 2016*). However, regulatable expression is often desired. Despite this, few regulatable systems have been reported and those that have are often reported to be leaky, resulting in low induction ratios (reviewed in: (*Fricke et al., 2021b*)). For example, the well-studied LacI system had low induction and high leakiness in the AAB *Gluconobacter* spp. and *K. xylinus* (*Condon, FitzGerald & O'Gara, 1991*; *Liu et al., 2020*). Similarly, the acyl homoserine lactone (AHL) inducible LuxR-$P_{lux}$ had high expression, but only 4-fold induction in free-living cells due to leakiness in uninduced *K. rhaeticus* (*Florea et al., 2016*). Recently two systems for regulatable gene expression in *Gluconobacter oxydans* have been reported, a L-arabinose-inducible AraC-$P_{BAD}$ system and an ATc-inducible Tn10-based TetR-dependent system. The L-arabinose-inducible AraC-$P_{BAD}$ system was reported to have low background leakiness and was inducible to 480-fold (*Fricke et al., 2020*). Interestingly, a similar system in *K. rhaeticus* was highly leaky and had low induction (*Teh et al., 2019*). While the Tn10-based TetR-dependent system was tightly regulated and is highly tunable in *G. oxydans*, a similar system was leaky and had low induction in *K. rhaeticus* (*Florea et al., 2016*; *Fricke et al., 2021b*). This highlights differences in control within the AAB despite their high evolutionary similarity (*Cleenwerck et al., 2009*; *De Vuyst et al., 2008*; *Papalexandratou et al., 2009*). Therefore, the response in each host must be confirmed prior to use.

To address the problem of low induction and leaky gene expression in AAB, we investigated the use of the LuxR-$P_{lux}$ system and a bicistronic TetR-dependent system for tightly regulated gene expression in *G. oxydans* using a pBBR1MCS-2 plasmid backbone. While both systems were regulatable, the amount of basal level expression varied between these systems. While the LuxR-$P_{lux}$ system showed strong induction, it lacked the ability for tunable expression of the bicistronic TetR system. These regulatory systems will add to the growing ability to control gene expression in AAB.

## MATERIALS AND METHODS

### Bacterial strains and culture conditions

*Gluconobacter oxydans* 621H (DSMZ 2343) was routinely grown at 30 °C and 250 rpm in yeast mannitol (YM) or yeast glucose (YG) medium (6 g/L yeast extract, 20 g/L mannitol or glucose, 2.5 g/L $MgSO_4 \times 7\ H_2O$, 1 g/L $(NH_4)_2SO_4$, 1 g/L $KH_2PO_4$, pH 6.0) with 50 µg/ml cefoxitin. *Escherichia coli* strains (Table 1) were grown in lysogeny broth (LB; 5g/L yeast extract, 10 g/L tryptone, 10 g/L sodium chloride) at 37 °C and 250 rpm. Agar was added to 1.5% when making solid medium. Kanamycin or ampicillin were added to 50 or 100 µg/ml for plasmid maintenance (Table 1). *G. oxydans* was transformed either by conjugation with *E. coli* S17-1 or by electroporation (*Kallnik et al., 2010*; *Kiefler, Bringer*

**Table 1 Strains, plasmids, and primers.**

| Strain | Description | Source |
|---|---|---|
| *G. oxydans* 621H | Wildtype | DSMZ 2343 |
| *E. coli* DH5α | *fhuA2 Δ(argF-lacZ)U169 phoA glnV44 Φ80 Δ(lacZ)M15 gyrA96 recA1 relA1 endA1 thi-1 hsdR17* | New England Biolabs |
| *E. coli* S17-1 | *recA pro hsdR RP4-2-Tc::Mu-Km::Tn7* integrated into the chromosome, *strR spcR tmpR*. | ATCC 47055 |
| *E. coli* 10β | *Δ(ara-leu) 7697 araD139 fhuA ΔlacX74 galK16 galE15 e14- φ80dlacZΔM15 recA1 relA1 endA1 nupG rpsL (StrR) rph spoT1 Δ(mrr-hsdRMS-mcrBC)* | New England Biolabs |
| *E. coli* JM109 | *endA1, recA1, gyrA96, thi, hsdR17 (rk−, mk+), relA1, supE44, Δ( lac-proAB), [F′ traD36, proAB, laqIqZΔM15]* | Promega |
| *Plasmids* | *Description* | *Source* |
| pBBR1MSC-2 | Broad-host-range derivative of pBBR1MCS; $Km^R$ | *Kovach et al. 1995*) |
| pBBR1p264-SP*pelB*-Streplong | pBBR1MCS-2 derivative containing the 5′-UTR of *gox0264*, an elongated C-terminal Streptag II and the signal sequence of *pelB*; $Kan^R$ | *Zeiser et al. 2014*) |
| pBBRp452-ST | pBBR1MCS-2 (*Kovach et al. 1995*) derivative containing the 5′-UTR of *gox0452* and a C-terminal Streptag II; $Kan^R$ | *Kallnik et al., 2010*) |
| pASK-IBA3 | Tetracycline inducible expression plasmid; $Amp^R$ | IBA GmBH |
| pUC57pTet | pUC57 containing the $P_{tet}$ promotor (iGEM BBa_R0040), RBS (iGEM BBa_B0034), the pASK-IBA3 MCS region, and the iGEM BBa_B0010 transcriptional terminator; $Amp^R$ | GenScript |
| p264TetR | pBBR1p264-SP*pelB*-Streplong derivative containing the TetR gene from pASK-IBA3 | This study |
| p452TetR | pBBR1p452-ST derivative containing the TetR gene from pASK-IBA3 | This study |
| p0169TetR | p452TetR derivative containing the *G. oxydans* p0169 promoter rather than the *G. oxydans* p452 promoter | This study |
| pTET0169-uidA | p0169TetR derivative containing the Tet regulon and MCS from pUC57pTet, and the *uidA* reporter gene | This study |
| pTET264-uidA | p264TetR derivative containing the Tet regulon and MCS from pUC57pTet, and the *uidA* reporter gene | This study |
| pTET452-uidA | p452TetR derivative containing the Tet regulon and MCS from pUC57pTet, and the *uidA* reporter gene | This study |
| p264LuxR | pBBR1p264-SP*pelB*-Streplong derivative containing the LuxR gene from *A. fischeri* | This study |
| p452LuxR | pBBR1p452-ST derivative containing the containing the LuxR gene from *A. fischeri* | This study |
| p0169LuxR | p452LuxR derivative containing the *G. oxydans* p0169 promoter rather than the *G. oxydans* p452 promoter | This study |
| pLUX0169-uidA | p0169LuxR derivative containing the $P_{lux}$ promotor (iGEM parts BBa_R0062), RBS (iGEM BBa_B0034), pASK-IBA3 MCS region, and the iGEM BBa_B0010 transcriptional terminator | This study |
| pLUX264-uidA | p264LuxR derivative containing the $P_{lux}$ promotor (iGEM parts BBa_R0062), RBS (iGEM BBa_B0034), pASK-IBA3 MCS region, and the iGEM BBa_B0010 transcriptional terminator | This study |
| pLUX452-uidA | p452LuxR derivative containing the $P_{lux}$ promotor (iGEM parts BBa_R0062), RBS (iGEM BBa_B0034), pASK-IBA3 MCS region, and the iGEM BBa_B0010 transcriptional terminator | This study |
| pUC57miniBicis | pUC57-mini derivative containing the Tetracycline-inducible bicistronic system from pZH512 (*Hensel, 2017*); $Amp^R$ | GenScript |
| pBICISTRON | pBBR1p452-ST containing the bicistronic tetracycline-inducible system | This study |
| *Primers* | *Sequence[a]* | *Endonuclease* |
| EcoRI/RBS/tetR_F | ATGAGAATTCAAAGAGGAGAAATACTAGATGTCTCGTTTAGATAAAAG | EcoRI |
| MluI/RBS/tetR_F | ATGAACGCGTAAAGAGGAGAAATACTAGATGTCTCGTTTAGATAAAAG | MluI |
| TetR_R | ATGAAAGCTTTTAAGACCCACTTTCACAT | HindIII |
| p0169_F | ATGCAGAGCTCTGAAAGCGGCTGGCGCGT | SacI |
| p0169_R | ATGCAGAATTCGCGGAAGGCGTTATACCCTGA | EcoRI |
| pTet_F | GCTCGAATGCCCCAGGGTC | PasI |
| pTet_R | CGAGCGCATTGTATACGAG | Bst1107I |
| BsaI_uidA_F | ATGGTAGGTCTCAAATGTTACGTCCTGTAGAAACCCCAAC | BsaI |

(Continued)

| Strain | Description | Source |
|---|---|---|
| BsaI_uidA_R | ATGGTAGGTCTCATATCATTGTTTGCCTCCCTGCTGCGG | BsaI |
| EcoRI/RBS/luxR_F | ATGAGAATTCAAAGAGGAGAAATACTAGATGAAAAACATAAATGCCGAC | EcoRI |
| MluI/RBS/luxR_F | ATGAACGCGTAAAGAGGAGAAATACTAGATGAAAAACATAAATGCCGAC | MluI |
| luxR_R | ATGAGGTCTCAAGCTGTTAATTTTTAAAGTATGGGC | BsaI |
| TetBicis_F | GTGCTCGAATGCGAGCTC | SacI |
| TetBicis_R | CACGAGCGCATTATTAATATAAAACG | VspI |

**Note:**
[a] Restriction endonuclease recognition site is underlined.

& Bott, 2017). Competent *E. coli* 10β and *E. coli* JM109 were prepared by the TSS method (Chung, Niemela & Miller, 1989).

## Materials and molecular techniques

Standard molecular techniques were done according to manufacturer's protocols or by standard protocols (Sambrook, Fritsch & Maniatis, 1989). The GeneJet Plasmid Miniprep kit (Thermo Fisher Scientific, Waltham, MA, USA) was used to purify plasmids and the GenElute Bacterial Genomic DNA kit (Millipore-Sigma, St. Louis, MO, USA) used to extracted genomic DNA. DNA was amplified using Phusion DNA polymerase and DreamTaq polymerase, and cloning done with FastDigest restriction enzymes and T4 ligase that were purchased from Thermo Fisher Scientific. Eurofins Genomics supplied all primers and sequencing services to confirm plasmids (Louisville, KY, USA) (Table 1).

## Plasmid construction

The *tetR* gene was amplified from pASK-IBA3 (IBA Lifesciences GmbH, Göttingen, Germany) using Phusion DNA polymerase and cloned into the MluI or EcoRI and HindIII sites of pBBR1p264-SPpelB-streplong and pBBR1p452-ST such that there were 6 bp between *tetR* and its RBS, creating p452TetR and p264TetR (Table 1). $P_{0169}$ was amplified from *G. oxydans* DNA and inserted into the SacI and EcoRI sites of p452TetR to create p0169TetR. The $P_{tet}$ promotor (iGEM BBa_R0040), ribosomal binding site (iGEM BBa_B0034), a MCS region containing both BsaI restriction sites from pASK-IBA3 (IBA Lifesciences GmbH, Göttingen, Germany), and the iGEM BBa_B0010 transcriptional terminator was synthesized and cloned into pUC57 by GenScript (Piscataway, NJ, USA) (Table 1). This $P_{tet}$-MCS-terminator region was amplified using pTet_F and pTet_R primers and inserted into the PasI and Bst1107I sites of the p0169TetR, p264TetR, and p452TetR plasmids. Additionally, the *uidA* gene encoding the β-D-glucuronidase from *E. coli* DH5α was amplified and inserted into the BsaI sites of the p0169TetR, p264TetR, and p452TetR to create pTET0169-uidA, pTET264-uidA, pTET452-uidA, having 7 bp between *uidA* and its RBS. All plasmids were confirmed by sequencing.

The *luxR* gene (iGEM BBa_C0062) was synthesized by GenScript, amplified with Phusion DNA polymerase, and inserted into the MluI or EcoRI and BsaI of pBBR1p264-SPpelB-streplong or pBBR1p452-ST such that there were 9 bp between *luxR* and its RBS,

creating p452LuxR and p264LuxR (Table 1). $P_{0169}$ was amplified from *G. oxydans* DNA and inserted into the SacI and EcoRI sites of p452LuxR to create p0169LuxR. The $P_{lux}$ promotor (iGEM BBa_R0062), ribosomal binding site (iGEM BBa_B0034), MCS region containing two BsaI restriction sites derived from pASK-IBA3 (IBA GmbH), and the iGEM BBa_B0010 transcriptional terminator was synthesized by Eurofins Genomics (Louisville, KY, USA). The synthesized fragment was cut with PasI and Bst1107I and ligated into similarly cut p0169LuxR, p264LuxR, and p452LuxR. The *uidA* gene encoding the β-D-glucuronidase from *E. coli* DH5α was also amplified and inserted into the BsaI sites of p0169LuxR, p264LuxR, and p452LuxR, having 10 bp between *uidA* and its RBS to create pLUX0169-uidA, pLUX264-uidA, pLUX452-uidA and confirmed by sequencing.

The tetracycline-inducible bicistronic expression system was constructed based on pZH512 plasmid that uses the iGEM BBa_R0040 $P_{tet}$ promoter, the *rrnB* iGEM BBa_B0010 terminator, and the GFPmut2 reporter (*Hensel, 2017*). The sequence derived from pZH512 was synthesized and cloned into pUC57-mini by GenScript (Piscataway, NJ, USA). The tetracycline-based bicistronic system was amplified using TetBicis_F/TetBicis_R primers and cloned into the VspI/SacI sites of pBBR1p452-ST such that there were 9 bp between *gfpmut2* and its RBS and 7 bp between *tetR* and its RBS, creating pBICISTRON.

### Reporter assay

Well grown cultures of *G. oxydans* were inoculated 1:50 in new 50 ml baffled shake flasks containing 10 ml YG or YM. Cultures were immediately induced by addition of either anhydrotetracycline (ATc) or N-(3-Oxohexanoyl)-L-homoserine lactone (AHL; Chemodex Ltd, St. Gallen, Switzerland) to the indicated final concentrations and grown to an $OD_{600}$ of about 0.8. The β-D-glucuronidase (UidA) activity was monitored in Miller Units as previously described (*Kallnik et al., 2010*). Cells were permeabilized by adding 30 μl cells to a 96 well plate containing 100 μl Z-buffer (60 mM $Na_2HPO_4 \times 7H_2O$, 40 mM $NaH_2PO_4 \times H_2O$, 10 mM KCl, 1 mM $MgSO_4 \times 7H_2O$, 3.5 mM 2-mercaptoethanol, 5% (v/v) chloroform, and 0.5% SDS, pH) and incubated for 10 min at 30 °C. Absorbance was monitored at 405 nm every minute on a BioTek EL808 plate reader after addition of 100 μl 4-nitrophenyl-β-D-glucuronide (4 mg/ml). All assays were done in at least three biological replicates each with three technical replicates and analyzed using BioTek Gen5 software.

Alternatively, expression was monitored by fluorescence using the GFPmut2 protein. Induced cultures were monitored for fluorescence in late log phase ($OD_{600}\sim0.8$). Fluorescence was measured with a λex of 485/4 nm and λem 508/4 nm using a SpectraMax M3 plate reader (Molecular Devices, San Jose, CA, USA). Fluorescence was measured in relative fluorescence units (RFU) and normalized to the optical density of the culture.

### Microscopy

*G. oxydans* pBICISTRON and wildtype were grown on YG to approximately 0.4 $OD_{600}$. Anhydrotetracycline was added to 100 ng/ml and cultures were incubated overnight. Clean depression slides were preheated in a 37 °C incubator. Molten buffered agarose (25 mM Tris-Base pH 7.4, agarose 1.2%) was applied to the depression slide and a glass

slide was placed on top to form an agar pad. The slide was removed once solidified and 5 µL of cells were placed onto the center of the pad. A Nikon Eclipse E600 fluorescence microscope (100×/1.40 numerical aperture oil immersion objective) with a Nikon DS-Qi2 camera. Fluorescence was measured using λex 460/40 nm / λem 510/50 nm filter. Digital images were acquired and analyzed with the NIS-Elements imaging software (version 5.20.01).

### Statistical analysis

R Studio was used to perform statistical analyses and to generate all plots (*R Core Team, 2021*). Analyses can be found in Script S1.

## RESULTS

### Regulation of the activation-dependent Luxr expression system in *G. oxydans*

The $P_{lux}$ promoter is a common system used to induce gene expression in many bacteria. The LuxR-$P_{lux}$ system was successfully used to control gene expression in *K. rhaeticus* (*Florea et al., 2016*). However, species and genera differences in regulation have been reported in the AAB (*Fricke et al., 2021a*; *Fricke et al., 2020*; *Fricke et al., 2021b*). Based on its reported high induction and with relatively lower leakiness in comparison to TetR-based expression control, a LuxR-$P_{lux}$ system was examined in *G. oxydans*. The *luxR* gene was cloned under the control of either the constitutive strong ($P_{0169}$ and $P_{264}$) or moderate ($P_{452}$) promoters (*Kallnik et al., 2010*; *Yuan et al., 2016*). Induction by addition of AHL was monitored with the *uidA* reporter gene under the control of the $P_{lux}$ promoter (Fig. 1). Regardless of the promoter used to express *luxR*, induction ratios were similar having average induction of 4.3 ± 0.31, 3.5 ± 0.17, and 4.90 ± 0.45 when $P_{0169}$, $P_{264}$, and $P_{452}$ promoters were used respectively (Fig. 2). Stronger absolute expression was observed when the $P_{0169}$ promoter was used to control *luxR* gene expression. However, induction ratios were similar to $P_{264}$, and $P_{452}$ despite the higher absolute activity. However, the amount of AHL inducer from 0.1 to 10 µM did not influence induction regardless of the strength of the promoter used to express the *luxR* gene. As such, this system functions as an on-off switch in the AHL concentration range tested rather than a tunable induction system in *G. oxydans*. Importantly, the LuxR-$P_{lux}$ system has previously been reported as inducible with lower leakiness in the acetic acid bacterium *K. rhaeticus* (*Florea et al., 2016*). However, the condition-dependent induction was only 5-fold in free-living cells and 12-fold in cellulose pellicles. The moderate levels of induction are attributed to the leakiness of this system in AAB. Indeed, similar induction levels are observed in *G. oxydans* and are also attributed to the moderate leakiness of the induction system. These data suggests that while the LuxR-$P_{lux}$ system shows good induction for heterologous gene expression relative to previously described systems in acetic acid bacteria (*Florea et al., 2016*), it is limited by its functionality as an on/off switch. Furthermore, tight control of gene expression is not possible due to the high background expression when not induced.
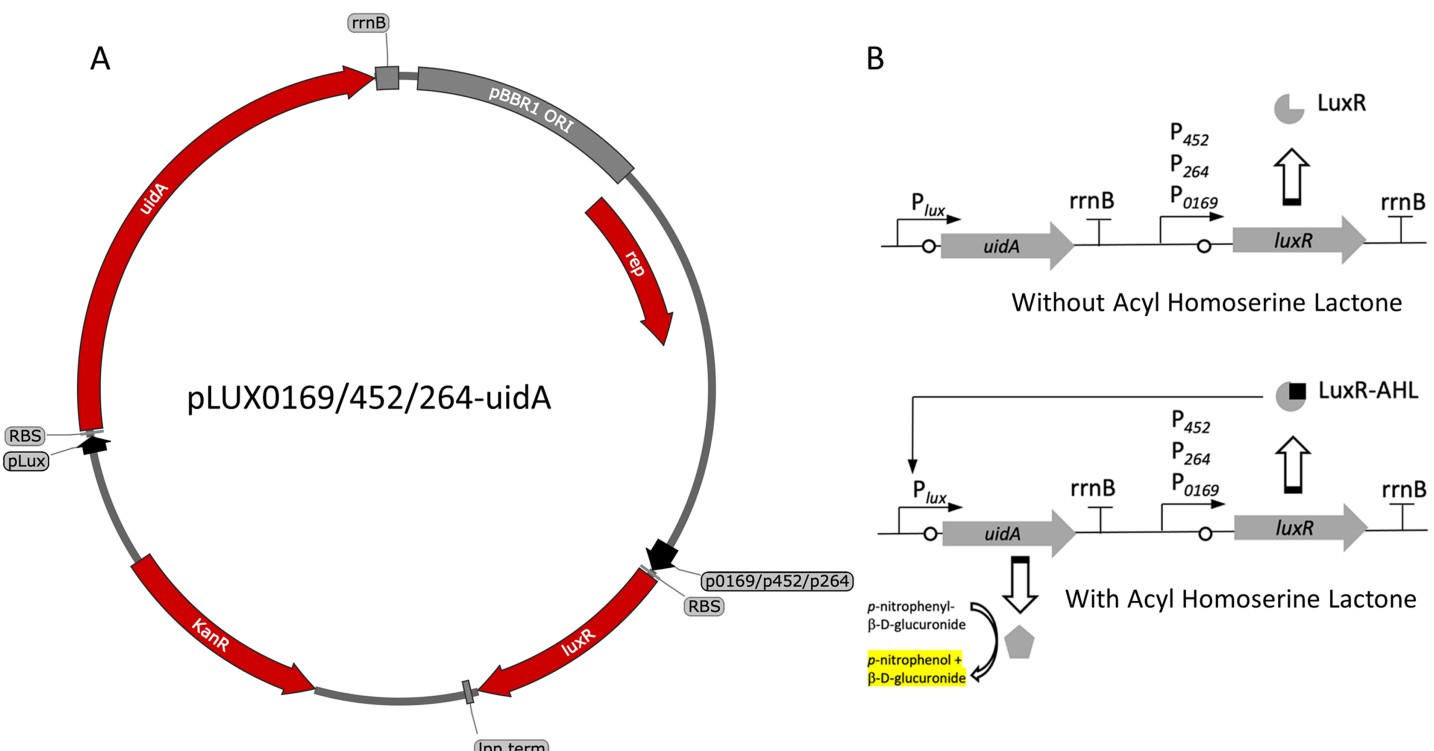

**Figure 1 Variants and regulation of the pBBR1MCS-2-based LuxR-P_lux expression system with a UidA reporter gene.** (A) Plasmid map of pLUX0169-uidA, pLUX452-uidA, and pLUX264-uidA. LuxR is derived from *A. fischeri* iGEM part BBa_C0062 and is constitutively transcribed from either strong promoters $P_{0169}$ or $P_{264}$ or the moderate $P_{452}$ promoter (*Kallnik et al., 2010*; *Shi et al., 2014*). LuxR is terminated by the *E. coli lpp* terminator (*Nishi et al., 1984*). The *uidA* reporter gene is expressed from the Lux promoter/operator region from iGEM BBa_0062 and is terminated by the *rrnB* terminator from iGEM BBa_B0010. The strong ribosomal binding site from iGEM BBa_B0034 is used for translation of both LuxR and UidA (*Hentschel et al., 2013*). (B) Regulation of the activation-dependent LuxR-P_lux expression system. Binding of AHL allows binding of LuxR-AHL to the P_lux promoter. UidA hydrolyzes pNPG to produce a yellow color that can be quantified colorimetrically.

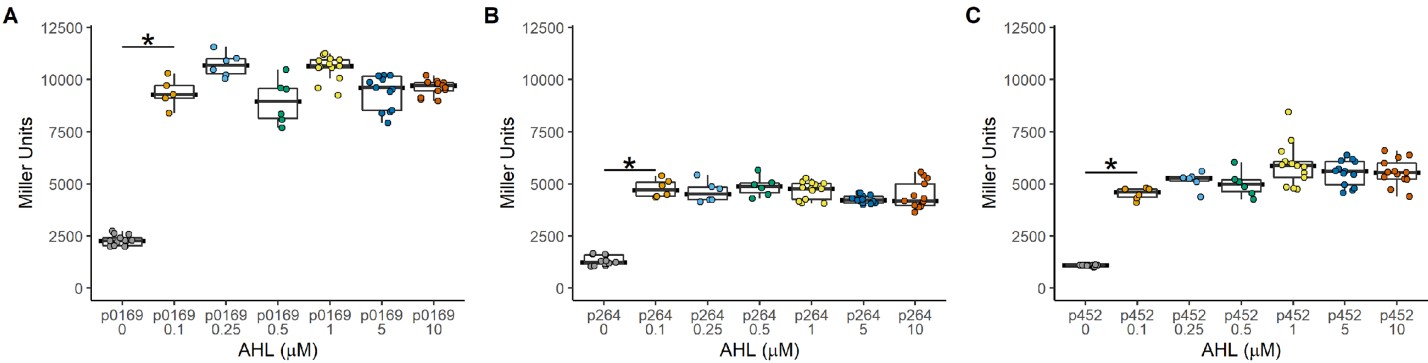

**Figure 2 Activity of the LuxR-P_lux systems in *G. oxydans*.** Induction was constant regardless of the amount of inducer and effectively functions as an on-off switch using the concentrations tested. (A) The PLUX0169-*uidA* using the strong $P_{0169}$ promoter to express the *luxR* gene had an induction ratio of 4.3 ± 0.31 (B) pLUX264-uidA using the strong $P_{264}$ promoter to express the *luxR* gene had an induction ratio of 3.5 ± 0.17 (C) pLUX452-uidA using the moderate $P_{452}$ promoter to express the *luxR* gene had an induction ratio of 4.90 ± 0.45. Data represent at least three biological replicates each with three technical repeats. An asterisk (*) indicates that the Kruskal-Wallis test had a $p < 0.001$.

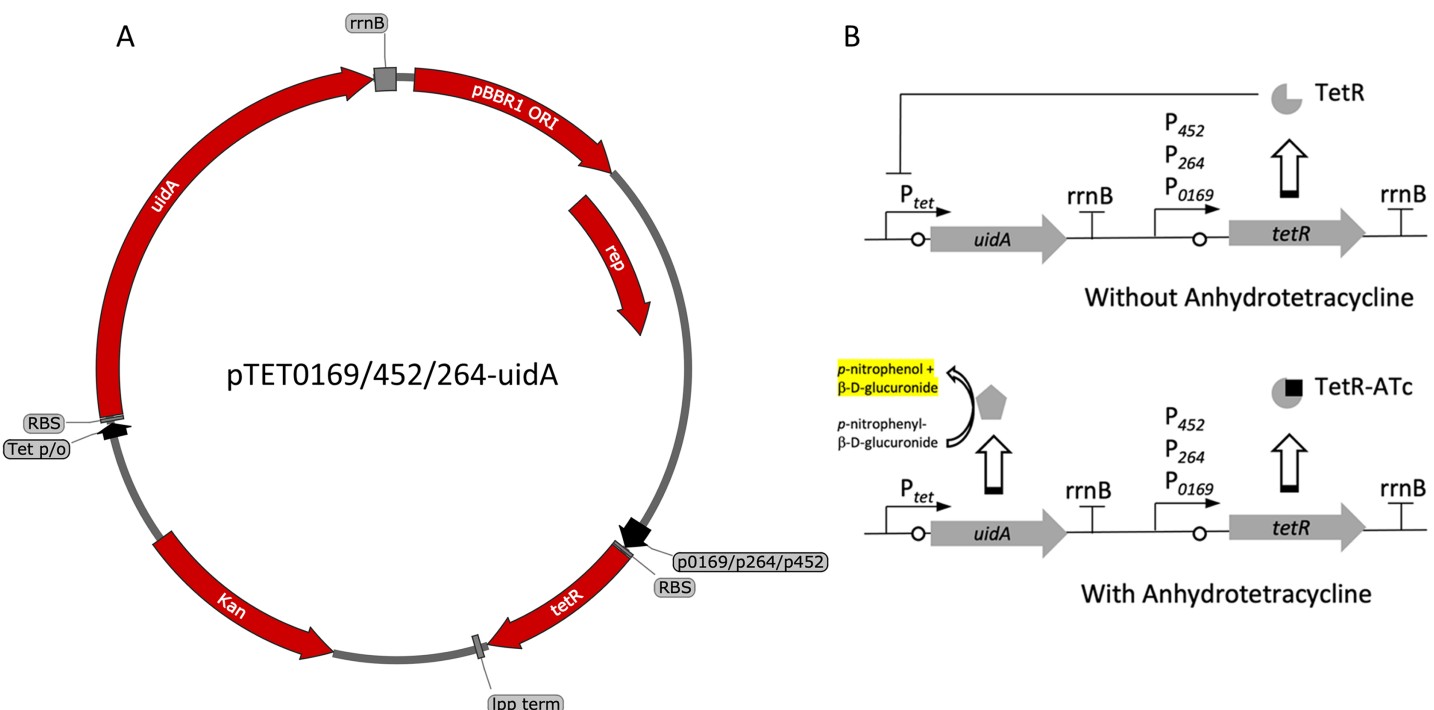

**Figure 3 Variants and regulation of the pBBR1MCS-2-based TetR-P$_{tet}$ expression system with a UidA reporter gene.** (A) Plasmid map of pTET0169-uidA, pTET452-uidA, and pTET264-uidA. The *tetR* gene is derived from the pASK-IBA3 plasmid (IBA Lifesciences GmbH, Göttingen, Germany). It is constitutively transcribed from either strong promoters P$_{0169}$ or P$_{264}$ or the moderate P$_{452}$ promoter (*Kallnik et al., 2010*; *Shi et al., 2014*). An *E. coli lpp* terminator was used to terminate *tetR* transcription (*Nishi et al., 1984*). The *uidA* reporter gene is expressed from the Tet promoter/operator region from iGEM BBa_0040 and is terminated by the *rrnB* terminator from iGEM BBa_B0010. The strong ribosomal binding site from iGEM BBa_B0034 is used for translation of both *tetR* and *uidA* (*Hentschel et al., 2013*). (B) Regulation of the constitutive repression TetR-P$_{tet}$ expression system. TetR binds ATc, relieving repression of the UidA reporter, which hydrolyzes pNPG, producing a yellow color that can be quantified colorimetrically.

## Regulation of the constitutive repression TetR expression system in *G. oxydans*

Another common system used to control heterologous gene express in bacteria is the TetR-P$_{tet}$ system. This system is a constitutive repression system, where TetR constitutively represses transcription of the target gene by binding the P$_{tet}$ promoter in the absence of a TetR effector. This system was examined for functionality in *K. rhaeticus* where gene expression was hindered by high leakiness of the repressor resulting in only *circa* 1.5-fold induction (*Florea et al., 2016*). However, variable expression in different AAB hosts has been previously observed, highlighting the need to examine expression in different hosts (*Teh et al., 2019*). To investigate its functionality in *G. oxydans*, a TetR-P$_{tet}$ system was examined. The *tetR* gene was cloned under the control of the constitutive P$_{0169}$, P$_{264}$, and P$_{452}$ promoters (*Kallnik et al., 2010*; *Shi et al., 2014*). Induction was monitored by addition of ATc with the *uidA* reporter gene under the control of the P$_{tet}$ promoter derived from Tn10 (Fig. 3). The TetR-P$_{tet}$ had strong maximal induction of 19-fold, 18.6-fold, and 4-fold when P$_{0169}$, P$_{264}$, and P$_{452}$ promoters were used to control *tetR* transcription respectively (Fig. 4). While promoters P$_{0169}$ and P$_{264}$ had similar induction ratios, absolute induction was 6-fold higher when the P$_{0169}$ promoter was used. This disparity in absolute activity

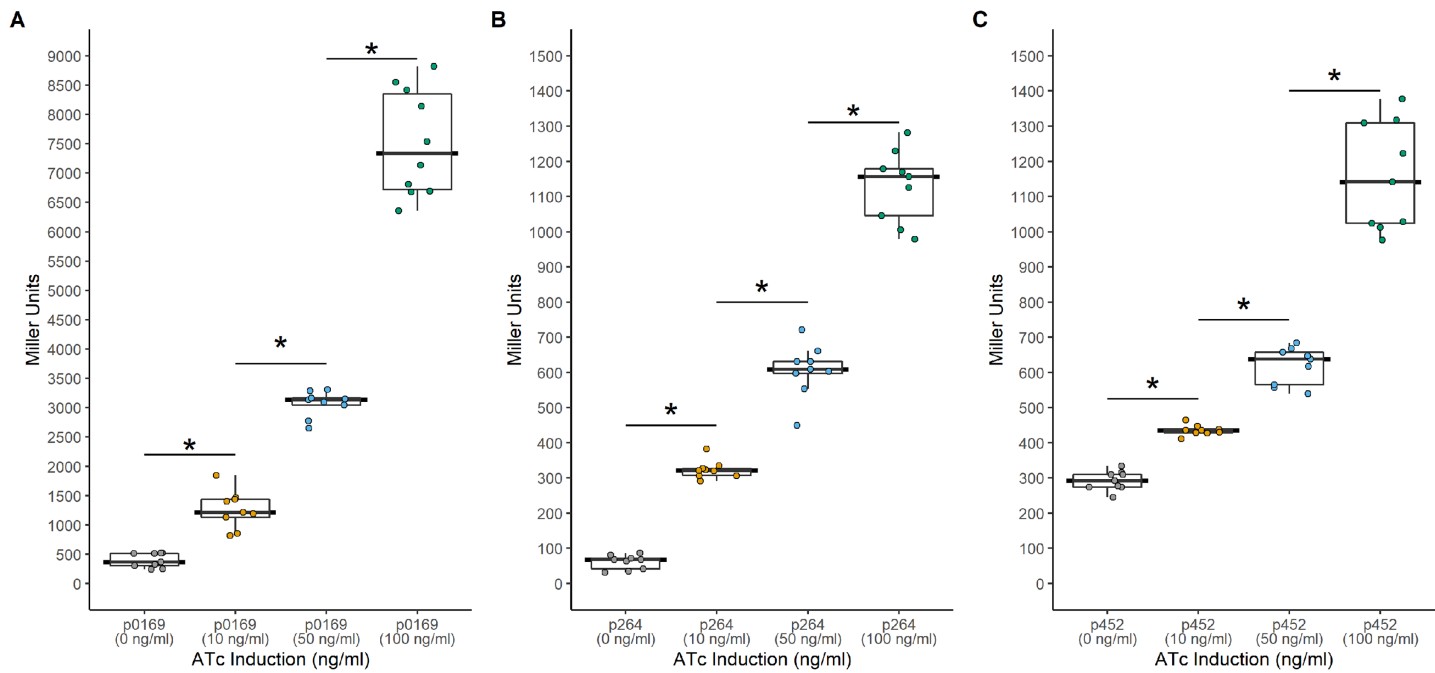

**Figure 4 Activity of the TetR-P$_{tet}$ systems in *G. oxydans*.** Background expression was low and induction was tunable with all three promoters. (A) The PTET169-uidA using the strong P$_{0169}$ promoter to express the *luxR* gene had an induction ratio of 19-fold (B) pTET264-uidA using the strong P$_{264}$ promoter to express the *luxR* gene had an induction ratio of 18.6-fold (C) pTET452-uidA using the moderate P$_{452}$ promoter to express the *luxR* gene had an induction ratio of 4-fold. Data represent at least three biological replicates each with three technical repeats. An asterisk (*) indicates that the Kruskal-Wallis test had a $p < 0.001$.

compared to induction ratio is attributed to the somewhat higher background expression when the P$_{0169}$ promoter was used. Interestingly, the induction ratio of P$_{452}$ was the lowest at 4-fold. This is likely due to the weaker expression of the TetR repressor. The lower amount of TetR in the cytoplasm likely is not sufficient to fully repress the P$_{tet}$ promoter, resulting in the increased amount of leaky expression that leads to reduced induction ratios. Indeed, the absolute levels of induction when the P$_{264}$ and P$_{452}$ promoter was used are similar, suggesting that the differences in induction ratios is driven by differences in promoter strength. However, induction of the TetR-P$_{tet}$ system using the P$_{452}$ promoter to express *tetR* is similar to induction with the LuxR-P$_{lux}$ system in *G. oxydans* and in *K. rheaticus* (*Florea et al., 2016*). Notably, the TetR-P$_{tet}$ system showed tunable expression in all cases, showing an increased response to increasing ATc concentrations in the concentration range examined. Interestingly, the TetR-P$_{tet}$ system showed stronger induction than the LuxR-P$_{lux}$ system in *G. oxydans*, which differs from observations in *K. rhaeticus* (*Florea et al., 2016*). However, the TetR-P$_{tet}$ system still had low levels of leaky expression when uninduced, limiting induction ratios.

## Construction of a tunable bicistronic TetR expression system in *G. oxydans*

While the above described TetR-P$_{tet}$ system showed strong induction and tunability, a low level of leakiness was still observed. This is similar to previously reported results of high basal level of expression of the Tet system in acetic acid bacteria in the absence of inducer
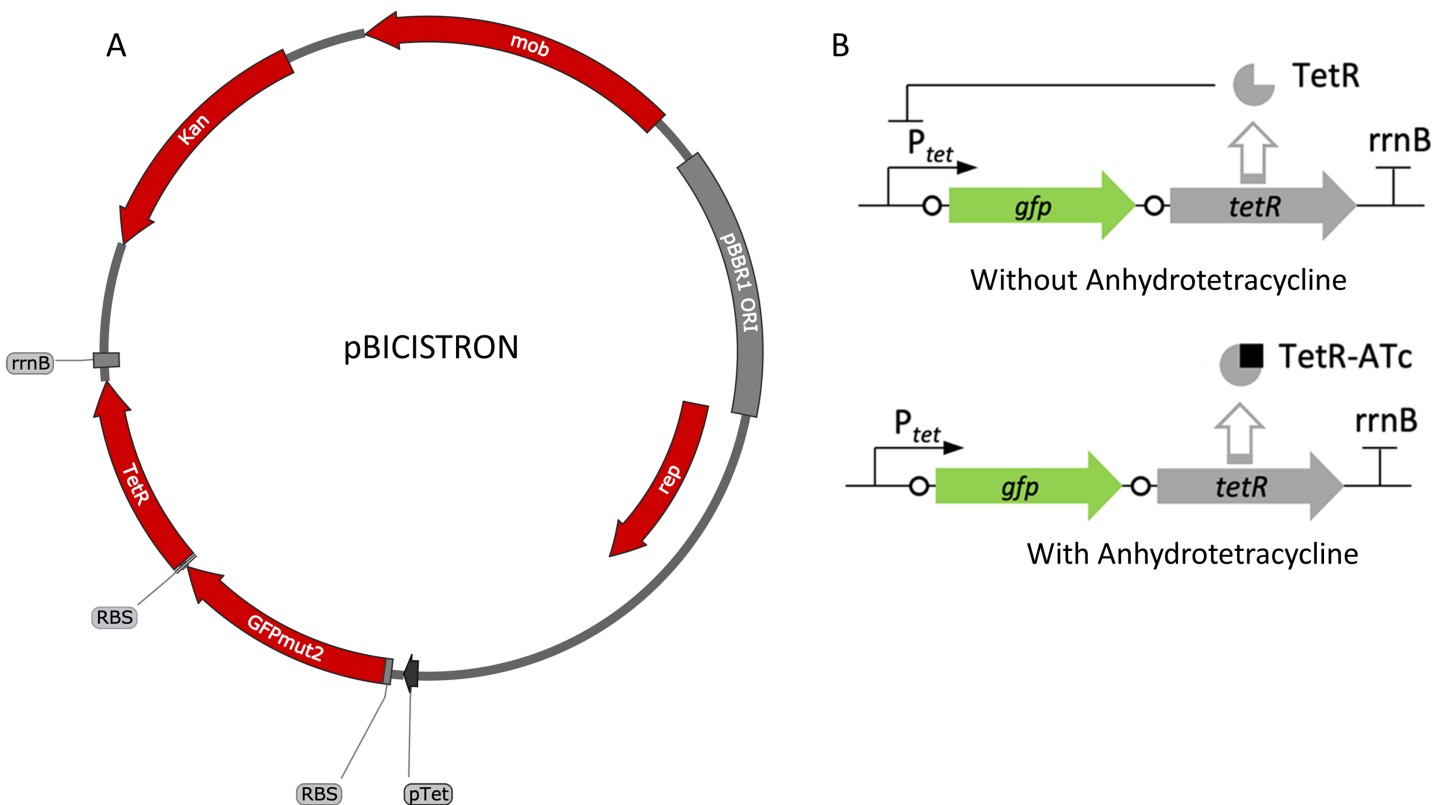

**Figure 5 Map and regulation of the bicistronic TetR-based autoregulation system.** (A) Plasmid map of pBICISTRON. The bicistronic gene arrangement was synthesized GenScript and is based on pZH512 (*Hensel, 2017*). These genes are transcribed by the $P_{tet}$ derived from iGEM BBa_R0040 and are terminated by the *rrnB* terminator from iGEM BBa_B0010. (B) Regulation of the bicistronic autoregulated expression system. TetR binds ATc, relieving repression of both *gfp* and *tetR*.

(*Florea et al., 2016*; *Fricke et al., 2021a*). In *E. coli*, a bicistronic autoregulation $P_{tet}$ system was predicted to have the lowest background noise and predicted to be tunable based on modeling of various gene configurations. These predictions were also validated experimental to show the optimal signal-to-noise ratio of a bicistronic gene arrangement (*Hensel, 2017*). We used this as a model to construct a bicistronic TetR system using a GFP reporter using the moderate strength RBS (aaagCCgagaaaggtaccgcATG, *Hensel, 2017*) in the 5′-UTR of the *gfpmut2* gene (Fig. 5). Like the TetR-$P_{tet}$ system, the bicistronic system showed a tunable response up to 200 ng/ml ATc addition (Fig. 6). The maximum GFP induction ratio was a 3,037-fold increase in fluorescence. Importantly, the bicistronic system was tightly controlled, having little to no fluorescence in the absence of inducer and signal only slightly above background levels of wildtype cells (Fig. 6). This is an approximately 160-fold improvement of induction compared to the TetR-$P_{tet}$ constitutive repression system in *G. oxydans* (Figs. 3 and 4). This is attributed to the low leakiness of the bicistronic system that increases induction ratios dramatically. Consequently, the bicistronic TetR gene arrangement is a promising system for gene control in *G. oxydans* and other AAB.

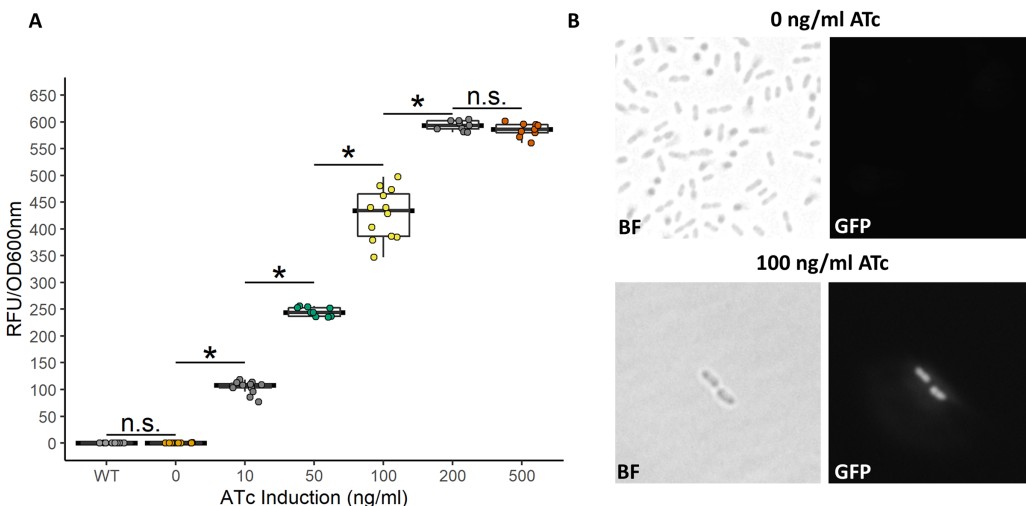

**Figure 6  Activity of the bicistronic TetR-based autoregulation system.** (A) The bicistronic TetR-based system had high induction ratios of 3,037-fold with low background expression. Data represent at least three biological replicates each with three technical repeats. An asterisk (*) indicates that the Kruskal-Wallis test had a $p < 0.001$. n.s., not significant. (B) Brightfield (BF) and fluorescence microscopy of uninduced and induced *G. oxydans* pBICISTRON. GFP was not detected when ATc was absent and was robust when induced (100 ng/ml ATc induction shown). A Nikon Eclipse E600 fluorescence microscope (100×/1.40 numerical aperture oil immersion objective) with a Nikon DS-Qi2 camera. Fluorescence was measured using GFP filter ($\lambda$ex 460/40 nm / $\lambda$em 510/50 nm). Digital images were acquired and analyzed with NIS-Elements imaging software. Representative view is shown.

## DISCUSSION

We looked at the use of three regulatable expression systems in *G. oxydans*, one using the common LuxR-$P_{lux}$ system and two genetic arrangements of the TetR-$P_{tet}$ system. The absolute induction of the LuxR-$P_{lux}$ system was high in *G. oxydans*, having and average activity of 9,750, 4,600, and 5,300 Miller Units with the $P_{0169}$, $P_{264}$, and $P_{452}$ promoters respectively (Fig. 2). However, the induction ratio was only about 4-fold regardless of the constitutive promoter used to express the *luxR* gene. This is similar to the induction ratio of 5-fold reported in *K. rhaeticus* free-living cells (*Florea et al., 2016*). The low induction ratio maybe due to the presence of rare *G. oxydans* codons in *luxR* that lead to slower translation rates or inclusion body formation, which would decrease reporter activity and induction ratios. However, slower translation has also been shown to increase the ratio of folded protein relative to rapid translation using only common codons, which can increase total protein while decreasing the folding yield ratio (*Rodriguez et al., 2018*). The induction ratio was also limited because of the leakiness of the expression system. In *Aliivibrio fischeri*, AHL is produced by a synthase encoded by the *luxI* gene. The expression of the *luxI* gene, along with the other bioluminescence genes, is under the control of the activator, LuxR (*Fuqua, Winans & Greenberg, 1994*). Yet, basal levels of AHL are still produced. It may be that LuxR has weak binding affinity for $P_{lux}$ in the absence of AHL, causing leaky transcription in *G. oxydans*. In the absence of the *luxR* gene, UidA activity was not detected. Consequently, the difference in background expression is likely

due to the increase in basal level of expression of LuxR from the different constitutive promoters, with the strongest $P_{0169}$ promoter enhancing basal level expression due to the increased levels of LuxR. Furthermore, the presence of a GinI/GinR quorum sensing system has been reported to be present in most AAB (*Iida, Ohnishi & Horinouchi, 2008*). This system is Lux-like and responds to AHLs. However, homologs of GinI/GinR were not identified by blastp. Despite this, an unknown GinI-like AHL-synthase or other undescribed AHL-dependent regulatory system may be present in *G. oxydans*. This would produce low levels of AHLs that may interact with LuxR, causing basal levels of expression, which may also explain the increased leakiness when the strong $P_{0169}$ promoter was used. Another drawback to this system is that it is not tunable with the AHL concentrations tested. Rather it functions as an on/off switch with equal induction ratios when using 0.1 µM AHL up to 10 µM of AHL (Fig. 2). Regardless of these shortcomings, the LuxR-$P_{lux}$ system is a promising candidate to control gene expression in *G. oxydans*, especially when high absolute levels of expression are desired or the basal-level expression is not of concern.

Alternatively, the TetR-$P_{tet}$ system has much lower levels of leaky expression in *G. oxydans* and up to 19-fold induction, which is approximately four-times higher compared to the LuxR-$P_{lux}$ system. Additionally, the TetR-$P_{tet}$ system was tunable, unlike the LuxR-$P_{lux}$ system. Interestingly, a similar TetR system was examined in *K. rhaeticus* but only had induction ratios of about 1.5-fold, despite both systems having a similar genetic arrangement (*Florea et al., 2016*). One difference in genetic arrangement is the presence of an intervening neo/kan cassette between the *tetR* gene and the $P_{tet}$ promoter (Fig. 3). It was suggested that terminator read-through may cause high leakiness of the $P_{tet}$ promoter in *K. rhaeticus* (*Fricke et al., 2021b*). Molecular simulations also showed that TetR dimerization is inhibited at low pH (*Fricke et al., 2021b*). *K. rhaeticus* is often grown in medium containing glucose or sucrose with an initial pH of 6.0, which decreases during growth (*Florea et al., 2016*; *Hestrin & Schramm, 1954*). The cytoplasmic pH of acetic acid bacteria can decrease to about 4.0 during growth (*Menzel & Gottschalk, 1985*), which may be responsible for the leakiness seen in high acid producing AAB like *K. rhaeticus* (*Fricke et al., 2021b*). Additionally, the separation of the *tetR* gene and the $P_{tet}$ promoter in our *G. oxydans* expression system likely preclude terminator read-through as the cause of the leaky expression. Instead, the leakiness may be attributed to the choice of promoter controlling the transcription of *tetR*. *K. rhaeticus* contains the $P_{J23118}$ or $P_{lacI}$ promoters for expression of *tetR* rather than native host constitutive promoters (*Florea et al., 2016*). It may be that the choice of promoter is important for sufficient expression of TetR and the repression of the $P_{tet}$ promoter, causing a tighter control of gene expression. Additionally, the promoter must not produce excessive TetR to avoid inclusion bodies that could reduce the concentration of functional TetR, resulting in leakiness. Indeed, the highest amount of uninduced leakiness was observed when the weaker $P_{452}$ promoter controlled *tetR* expression (Fig. 4).

In an attempt to reduce the amount of leakiness and improve induction of the TetR-$P_{tet}$ system, we engineered a bicistronic TetR expression system. This gene arrangement has the best signal-to-noise ratio of any gene arrangement for the TetR-controlled expression

in *E. coli* (*Hensel, 2017*). In *G. oxydans*, this system exhibited very low uninduced expression with high induction ratios of >3,000-fold. Importantly, the system was highly tunable when up to 200 ng/ml of ATc was added. Recently, another TetR-based system was reported in *G. oxydans*. This system uses the native Tn10 gene orientation. This Tn10-based system was also tunable and had induction ratios up to 3,674-fold (*Fricke et al., 2021b*), which is similar to the bicistronic system reported here. Tunability is important in many instances. For example, membrane-bound dehydrogenases in AAB often require low-to-moderate gene expression for functional enzyme production (*Mientus et al., 2017*). Also, in some cases strong gene expression is known to produce inclusion bodies. Metabolic and genetic engineering often requires that genes are expressed at lower levels. Therefore, the ability to tune gene expression up or down as needed is essential in these applications. In these cases, the bicistronic or Tn10-based TetR-dependent systems may be preferred over the LuxR system that is not tunable and shows higher basal levels of uninduced expression.

## CONCLUSIONS

It recently has become possible to control and tune gene expression in AAB and there are now multiple systems for gene control. In addition to the LuxR-$P_{lux}$ systems described here and in *K. rhaeticus*, there are two known tunable and highly inducible TetR-$P_{tet}$ systems (*Florea et al., 2016*; *Fricke et al., 2021b*). Another promising system for regulatable gene expression in AAB is the AraC-$P_{BAD}$ system. This system had low leakiness and is inducible up to 480-fold in *G. oxydans*. Importantly, this system was also tunable by varying the amount of arabinose added to the medium (*Fricke et al., 2020*). The current development of multiple tunable expression systems in AAB is advantageous for future studies into not only basic biological questions, but also for engineering of new strains for industrial productions.

## ACKNOWLEDGEMENTS

We are grateful to Drs. Bonnie Bratina, Todd Osmundson, and Dan Bretl for helpful discussions and feedback during the experimentation and writing of this paper.

### Funding
This project was funded by the University of Wisconsin–La Crosse through startup funds and a Undergraduate Research and Creativity Grant. The funders had no role in study design, data collection and analysis, decision to publish, or preparation of the manuscript.

### Grant Disclosures
The following grant information was disclosed by the authors:
University of Wisconsin–La Crosse.

### Competing Interests
The authors declare that they have no competing interests.
## Author Contributions

- Monica Bertucci performed the experiments, analyzed the data, prepared figures and/or tables, authored or reviewed drafts of the article, and approved the final draft.
- Ky Ariano performed the experiments, analyzed the data, prepared figures and/or tables, authored or reviewed drafts of the article, and approved the final draft.
- Meg Zumsteg performed the experiments, authored or reviewed drafts of the article, and approved the final draft.
- Paul Schweiger conceived and designed the experiments, performed the experiments, analyzed the data, prepared figures and/or tables, authored or reviewed drafts of the article, and approved the final draft.

## Data Availability

The script used for statistical analysis and figure generation are available in the Supplemental File.

## Supplemental Information

Supplemental information for this article can be found online at http://dx.doi.org/10.7717/peerj.13639#supplemental-information.

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
