# Peer review of "Engineering a tunable bicistronic TetR autoregulation expression system in Gluconobacter oxydans"

_PeerJ, doi:10.7717/peerj.13639_

## Round 0.1 · original submission · Major Revisions

Dear Dr. Bertucci and colleagues:

Thanks for submitting your manuscript to PeerJ. I have now received two independent reviews of your work, and as you will see, the reviewers raised some concerns about the research. Despite this, these reviewers are optimistic about your work and the potential impact it will have on research studying metabolic engineering of acetic acid bacteria. Thus, I encourage you to revise your manuscript, accordingly, taking into account all of the concerns raised by both reviewers.

While the concerns of the reviewers are relatively minor, this is a major revision to ensure that the original reviewers have a chance to evaluate your responses to their concerns. There are many suggestions, which I am sure will greatly improve your manuscript once addressed.

Importantly, please ensure that your figures and tables contain all of the information that is necessary to support your findings and observations.

I look forward to seeing your revision, and thanks again for submitting your work to PeerJ.

Good luck with your revision,

-joe

Reviewer 1 ·

Basic reporting

Overall, the manuscript is written in clear, unambiguous, professional English language. The introduction and background show the context. Relevant literature is well referenced. The figures are relevant, well labelled and described.

Experimental design

According to the AAB literature available, the experiments and data presented are original primary research and are within the scope of the journal. The research question is well defined, relevant and meaningful. The experiments have been performed according to technical standards. The methods are described with sufficient detail and information to replicate.

Validity of the findings

The data presented appear to be robust and statistically sound. The conclusions are stated, linked to the author’s original research question and mainly limited to the supporting results. The conclusion on the LuxR-Plux system seems to be too early without further results.

Additional comments

Comments on the manuscript “Engineering a tunable bicistronic TetR expression system in Gluconobacter oxydans“

The availability of genetic systems for tunable expression of target genes in acetic acid bacteria (AAB) is still very limited. One reason for this is the (high) leakiness of classical repressor-based expression systems often seen in AAB. Now, the manuscript presented adds an interesting aspect for engineering tunable expression systems for AAB by pointing to the E.coli study of Hensel (2017), in which stochastic simulation showed that negative autoregulation of bicistronic expression of the repressor gene and a gene of interest can be necessary to reduce noise and produce more gradual response to induction. This has not yet been tested in AAB. Consequently, the authors tested this design principle based on autoregulation in the AAB G. oxydans. For a TetR-Ptet system they found that the bicistronic tetR expression system constructed and tested was inducible up to more than 3000-fold and was highly tunable with almost no background expression when non-induced. This makes bicistronic tetR systems potentially useful for engineering G. oxydans and possibly other AAB.

The following comments are intended to improve the manuscript:

The aspect of “autoregulation“ is suggested to be included already in the title.

Lines 199-201 and Figure 2:
Why is non-induced UidA reporter activity of p452-luxR twice as high compared to the other constructs? Plux-uidA is always the same in all constructs and basal expression should always be the same? Could this be related to luxR expression levels and discussion lines 269-274?
Using the weak P452 for luxR expression is resulting in the highest induced UidA activities, while induced UidA activities were much lower when luxR was expressed with strong P0169 or P264.
How about inclusions bodies resulting in non-functional LuxR protein?
Furthermore, how about codon usage of the luxR sequence in G. oxydans? Are there some/many rare codons worth to be mentioned in text?

Line 210:
5-fold to 12-fold increase is good induction? How do the authors define good induction?

Line 212:
“... low background expression when not induced“? Wasn‘t it a higher non-induced background expression?

Lines 215-216:
"when not induced" or "in the absence of an TetR effector" should be added.

Line 237:
A sigmoidal response cannot be seen in Figure 4.

Lines 256-257:
Here, also name your system and refer to the system from Figures 3+4.

Line 263:
Contradiction: highly induced, yet only 5-fold? What is the author´s definition of highly induced?

Lines 269-270:
What is the expression strength of Plux in G.oxydans in the abscence of the luxR gene?

Lines 272-274:
What are the top 2 candidates in G.oxydans according to the BLAST searches?

Lines 275-277 and Line 318:
First, lower levels of AHL (gradually below 1 µM) should be tested and data should be shown, before this conclusion can be drawn.

Lines 278-279:
Confusing: the same data sometimes are considered as low levels of basal expression and sometimes as high levels of basal expression?

Line 297:
According to the author‘s discussion in the lines below, the point is not "directing the transcription" per se, rather it is "driving/controlling the transcription". I think "Directing" would be misleading here.

Line 299:
Not only sufficient expression, also not too high expression to avoid inclusion bodies / non-functional TetR protein. Also check wording (“the repression the“).

Lines 102-104 and elsewhere:
put always a space between number and unit

Lines 118-151 plasmid construction:
The spacings between the RBS and the start codons should be mentioned in the text.

Figure 3 legend:
- UidA should be uidA and italic
- tetR gene (not TetR) is terminated
- translation of tetR and uidA, not TetR nor UidA

Line 209: check wording
Line 225: check wording (“in when“)
Line 231: check wording (“in in“)
Line 234: check wording
Line 236: check commas
Line 253: check wording
Line 291: check wording / grammar / typo
Line 315: check wording

Please explain n.s. in Figure 6.

Reviewer 2 ·

Basic reporting

The paper is well written.

Experimental design

This work is well designed.

Validity of the findings

The findings are valuable in this field.

Additional comments

Tunable expression of desired protein is a key technology for protein engineering, metabolic engineering, studying physiology, biochemistry, and so on. Bertucci et al. developed such the expression systems in their manuscript entitled “Engineering a tunable bicistronic TetR expression system in Gluconobacter oxydans”. Honestly, I do not appreciate this paper as for novelty. Also, it is confusing that use of the reporter gene are different in the experiments. Even though the criticism is true, the biological experiments are well and convincing, and the paper is well written. Thus, I recommend this manuscript to accept for publication, if they consider the following suggestions.

Major concern:
1. Interpretation on the results of LuxR is strange for me. I anticipate a simple possibility that the affinity of LuxR to AHL is higher than they expected. The dissociation constant of LuxR with AHL would be less than 1 micro M. Do the authors have any evidence to exclude this possibility?
2. Do the authors have any experimental results to connect Figures 4 and 6? Because the reporter genes used in these experiments are different from each other. Therefore, it is difficult to compare two data. Alternatively, please explain a compatibility of two experiments, i.e. Figures 4 and 6.

Minor concern:
1. What about growth behaviors of the recombinant G. oxydans strains constructed in this study?
2. L250, What is “medium strength RBS”? I understand ribosome-binding sequence (site), but what is medium strength? One or more citations may helpful, if available.
3. L253, “The maximal GFP induction ratio” or “The maximum GFP induction ratio”?
4. L317, “Tn10-based”.
5. Table 1, Please italicize genotype.
6. Table 1, I recommend to insert a new line for pBBR1MCS-2.

---

## Round 0.2 · accepted · Accept

Dear Dr. Bertucci and colleagues:

Thanks for revising your manuscript based on the concerns raised by the reviewers. I now believe that your manuscript is suitable for publication. Congratulations! I look forward to seeing this work in print, and I anticipate it being an important resource for groups studying metabolic engineering of acetic acid bacteria. Thanks again for choosing PeerJ to publish such important work.

Best,

-joe

Reviewer 1 ·

Basic reporting

no comment

Experimental design

no comment

Validity of the findings

no comment